# Repolarization of Unbalanced Macrophages: Unmet Medical Need in Chronic Inflammation and Cancer

**DOI:** 10.3390/ijms23031496

**Published:** 2022-01-28

**Authors:** Yannick Degboé, Rémy Poupot, Mary Poupot

**Affiliations:** 1Infinity, Université Toulouse, CNRS, INSERM, UPS, 31024 Toulouse, France; yannick.degboe@inserm.fr; 2Département de Rhumatologie, CHU Toulouse, 31029 Toulouse, France; 3Centre de Recherche en Cancérologie de Toulouse, Université Toulouse, INSERM, UPS, 31037 Toulouse, France; mary.poupot@inserm.fr

**Keywords:** cancer, dendrimer, macrophages, monocyte, osteoclast, polarization, rheumatoid arthritis, tumor-associated macrophages

## Abstract

Monocytes and their tissue counterpart macrophages (MP) constitute the front line of the immune system. Indeed, they are able to rapidly and efficiently detect both external and internal danger signals, thereby activating the immune system to eradicate the disturbing biological, chemical, or physical agents. They are also in charge of the control of the immune response and account for the repair of the damaged tissues, eventually restoring tissue homeostasis. The balance between these dual activities must be thoroughly controlled in space and time. Any sustained unbalanced response of MP leads to pathological disorders, such as chronic inflammation, or favors cancer development and progression. In this review, we take advantage of our expertise in chronic inflammation, especially in rheumatoid arthritis, and in cancer, to highlight the pivotal role of MP in the physiopathology of these disorders and to emphasize the repolarization of unbalanced MP as a promising therapeutic strategy to control these diseases.

## 1. Introduction

Macrophages (MP), the tissue counterpart of monocytes circulating into the blood stream, are innate immune cells that have a pivotal role in inflammation, tissue development, and homeostasis. Their most described functions are phagocytosis [1], angiogenesis, and production of both pro-inflammatory and anti-inflammatory mediators. However, the functions of MP extend beyond innate immunity as they interface with adaptive immunity via both the secretion of cytokines and the presentation of antigens. The functions fulfilled by MP are tightly related to their capacity to sensor and integrate a wide panel of stimuli present in their environment. This recognition of multiple stimuli is made possible thanks to the expression of a variety of receptors: cytokine receptors, pattern-recognition receptors (PRR), and phagocytosis receptors.

### 1.1. Cytokine Receptors

MP express all six different families of receptors for innate immunity cytokines (Figure 1):-Hematopoietin (i.e., class I cytokines) receptors have a common chain that accounts for the ligand specificity (the γ-chain for interleukin-2 (IL2), the β-chain for the granulocyte macrophage-colony stimulating factor (GM-CSF)). They recognized GM-CSF and IL6, among others, and they signal via the JAK-STAT (Janus kinases-signal transducer and activator of transcription) pathway [2];-Interferon (i.e., class II cytokines) receptors are heterodimeric receptors that also signal via the JAK-STAT pathway [3];-Tumor necrosis factor (TNF) family receptors recognize TNF, and also Fas-ligand, CD40-ligand. The binding of the ligand triggers the trimerization of the receptor [4], leading to the activation of the trimer receptor at the membrane of MP. However, these receptors can be cleaved, and are therefore present as soluble receptors in the microenvironment. These receptors are characterized by a death domain (such as TRADD (tumor necrosis factor receptor type 1-associated death domain), and FADD (Fas-associated protein with death domain)) at their cytoplasmic tail, and signal due to adaptors associated therewith;-Immunoglobulins superfamily receptors recognize cytokines of the IL1 family (IL1α/β), the interleukin-1 receptor antagonist (IL1-ra), IL18, and IL33, and growth factors such as macrophage-colony stimulating factor (M-CSF) [5]. They are characterized by an “immunoglobulin-like” extracellular domain and signal after dimerization via Toll/interleukin-1 receptor (TIR) of their cytoplasmic tail, involving signaling proteins such as interleukin-1 receptor-associated kinase (IRAK) and myeloid differentiation primary response protein 88 (Myd88);-Transforming growth factor (TGF) receptors recognize TGFβ, among others. They have a serine/threonine kinase activity and signal via either a hetero-tetrameric complex that induces Smad complexes, or a Smad-independent pathway such as mitogen-activated protein kinases (MAPK), phosphoinositide 3 kinase (PI3K), and Rho-GTPases [6];-Chemokine and formyl-methionyl-leucyl-phenylalanine (fMLP) receptors recognize in particular IL8, C-C chemokine ligand 2 (CCL2) (also called monocyte chemoattractant protein 1 (MCP1)). These receptors harbor seven transmembrane domains and are coupled to hetero-trimeric G proteins that mediate the signaling cascade [7].

### 1.2. Pattern-Recognition Receptors (PRR)

PRR are expressed both at the surface of MP (membrane receptors) and in internal compartments (endosomes, cytosol). They recognize conserved patterns leading to a pro-inflammatory activation of MP. These conserved patters are either extrinsic danger signals (pathogen-associated molecular patterns, PAMP) or intrinsic danger signals (damage-associated molecular patterns, DAMP). There are three different types of PPR:-Toll-like receptors (TLR) recognize intracellular (TLR3, TLR7, TLR8, TLR9) or extracellular (TLR1, TLR2, TLR4, TLR5, TLR6) danger signals [8];-Nucleotide oligomerization domain (NOD)-like receptors (NLR), such as NOD-like receptor family pyrin domain containing 3 (NLRP3) and NOD1, recognize cytoplasmic patterns mainly of bacterial origin, and induce the formation of inflammasome [9];-Retinoic acid-inducible gene I-like receptors (RLR), such as melanoma differentiation-associated protein 5 (MDA5) and RIG-I, recognize cytoplasmic patterns from viral origin and induce an interferon (IFN) type 1 response [10].

### 1.3. Phagocytosis Receptors

They sensor stimuli from the environment of MP:-Scavenger receptors (such as CD36 and CD163) that in particular recognize modified lipoproteins [11]. Scavenger receptors are able to associate with co-receptors. This ability broadens the variety of ligands they recognize and the associated functions they carry out, including the clearance of pathogens, and the transport of lipids and cargos within the cells. Therefore, scavenger receptors are involved in the immune response, in particular in the polarization of MP, and thereby in the pathogeny of inflammatory disorders;-Lectin receptors such as mannose receptor (CD206), Dectin-1, and dendritic cell-specific intracellular adhesion molecule-3-grabbing non-integrin (DC-SIGN, also named CD209) that have a binding domain for carbohydrates [12]. Lectin receptors are crucial for adapting the immune response to pathogens. Their activation leads to the secretion of cytokines that will shape the immune response of T lymphocytes. This property can be advantageously taken into account to promote new vaccination approaches;-The γ receptors for the constant fragment (Fc) of immunoglobulins, such as CD16 (FcγRIII), CD32 (FcγRII), and CD64 (FcγRI), that bind immunoglobulins, in particular those from opsonized particles [13]. The recognition of immunoglobulins and immune complexes by activating FcγR influences the uptake processing, and presentation of antigens by monocyte-derived DC and MP, both in the steady state and during inflammation.

### 1.4. The Ontogeny of Macrophages

Resident MP are found in all tissues. For years, the prevalent dogma has stated that MP infiltrate tissue deriving from circulating monocytes, therefore with a post-natal hematopoietic origin. This dogma has been tackled in the 2010s when it was discovered that specific “resident” MP of the brain (namely, microglia [14]) and of epidermis (namely, Langherans cells [15]) originate from embryonic precursors. Importantly, the vast majority of published studies depict the ontogeny of mouse MP, but not that of human ones.

In mouse, embryonic hematopoiesis is divided in three distinct waves:(i)a primitive wave of erythro-myeloid precursors from the yolk sac at day E7.5. These cells express the receptor for M-CSF (CSF-1R), but not the transcription factor c-Myb;(ii)a second wave called “transient definitive” that comes from the hemogenic endothelium of the yolk sac. This wave generates erythromyeloid precursors (c-Myb^+^) that will migrate to the fetal liver thanks to the development of the blood vasculature at day E8.5;(iii)a third wave called “definitive” that also comes from the hemogenic endothelium of the yolk sac. It produces hematopoietic stem cells (c-Kit^+^ Sca-1^+^) at day E10.5 in aorta, gonads, and mesonephros. These precursors will set up the definitive hematopoiesis of both fetal liver and fetal bone marrow.

As of today, three mechanisms are identified for the generation and the maintenance of mouse MP (Figure 2) [16]:(i)In “closed” tissue, such as brain, skin, and lungs, “resident” MP originate from fetal hematopoiesis and are self-renewed. Of note, the particular status of the liver in which Kupfer cells may have a minor contribution of the neonate homeostasis;(ii)In open tissue with slow kinetics (heart, pancreas), MP originating from adult hematopoiesis rapidly replace the ones originating from embryonic hematopoiesis;(iii)In open tissue with fast kinetics (lamina propria, dermis), MP derive from circulating monocytes and depend constantly on blood for their renewal.

From a phenotypic point of view, mouse “resident” MP express F4/80, CD64, CD14, and MerTK, whereas infiltrating MP express high levels of CD11b, CCR2, and Ly6C.

### 1.5. The Polarization of Macrophages

The first description of the inflammatory MP within the context of bacterial infections can be traced back to 1962 [17]. Later on, the pivotal role of the interferon-γ (IFNγ) was proven to explain the anti-tumor and anti-bacterial abilities of these cells, via the production of reactive oxygen species (ROS) [18]. The paradigm of the alternative, anti-inflammatory activation of MP by IL4 was described in 1992 [19]. This anti-inflammatory phenotype is characterized by a spindle-shaped morphology, a high expression of CD206 (the mannose receptor), a higher capability for the clearance of mannosylated ligands, an increase of the expression of the major histocompatibility complex (MHC) class II molecules, and a reduced secretion of both superoxide anion and inflammatory cytokines (such as IL1, IL8, and TNF).

The concept of the M1/M2 polarization of MP was proposed at the beginning of the 2000s [20]. It is based on the observation that MP from Th1 and Th2 mice (such as C57BL/6 or B10/D2, and Balb/c or DBA/2, respectively) have a different metabolism of arginine. MP from Th1 mice produce more nitric oxide (NO) from arginine, whereas MP from Th2 mice preferentially metabolize arginine in ornithine. Over the years, the M1/M2 dichotomous polarization of MP was progressively refined with “intermediate” phenotypes (M2a, M2b, M2c, etc.) involved in immuno-regulation or in tissue repair [21,22,23,24,25]. Tumor associated macrophages (TAM) can be added to these phenotypes [26], and are part of this review. TAM can also be differentiated in M1 or M2 phenotypes depending on the cytokinic environment in the tumor micro-environment (TME). In the early stage of tumors, the production of IFNγ by Th1 cells induces an M1 phenotype for TAM with anti-tumor properties. When the TME becomes more pro-tumor, IL4, IL13, or IL10, produced by Th2 or Treg cells, promote M2 or M2-like phenotype for TAM, displaying protumor functions.

However, this classification system does not take into account the diversity of the stimuli activating MP in vivo, nor the physiological or the pathophysiological contexts. Moreover, the vast majority of the studies, of which, the ones cited above [21,22,23,24,25] were performed in mice, therefore their relevance to humans can be questioned [27]. The development of molecular characterization of MP, in particular transcriptomic, high throughput screening, and single cell technologies [28,29,30,31], have refined our knowledge about the dynamics of the polarization of MP. Taking into account the multidimensional feature of the polarization of MP, and the plasticity thereof, international experts have proposed that for in vitro studies, MP are named after the stimuli that are used (Figure 3) [32]. The figure shows that MP activated by IL4 are the most M2 cells of the spectrum, followed by MP activated by immune complexes (Ic), IL10, and finally MP activated by the combination of glucocorticoids (GC) + TGFβ, and GC alone. On the opposite side of the spectrum are MP activated by LPS, then LPS + IFNγ, and finally IFNγ, that display increasingly pronounced M1 phenotype.

## 2. Managing Macrophages in Inflammatory Disorders

As stated above, MP can drive as well as resolve inflammation in pathogenic conditions. The context of rheumatoid arthritis (RA) exemplifies this functional diversity and the impact of the therapeutic modulation of MP at cell and molecular levels.

### 2.1. Macrophages in Rheumatoid Arthritis

#### 2.1.1. Subpopulations of Synovial MP

Synovium is a complex environment composed of niches sited in two layers: the lining layer and the sub-lining layer. Normal synovium is composed of one to three layers of synoviocytes corresponding to fibroblast like synoviocytes (FLS) and resident MP [33].

Decades ago, histological studies of rheumatoid synovium identified the involvement of MP in arthritis. During this process, synovium undergoes massive changes including hyperplasia of synovial lining layer, major accumulation of CD68^+^ MP, and infiltration of immune cells [34]. Histology combined with RNA sequencing (RNA-seq) of synovial biopsies from early RA patients allowed defining of three different “pathotypes” of MP: lympho-myeloid (CD20 B cell aggregate rich), diffuse myeloid (CD68 rich in the lining or sub-lining layer but poor in B cells), and pauci-immune fibroid (paucity of immune-inflammatory cell infiltration) [35].

The historical view of synovial MP depicted two types thereof: the tissue-resident MP maintaining homeostasis, and the infiltrating MP derived from blood monocytes and involved in inflammation. The input of recent approaches enables refining of the classification of MP in synovium, especially in the context of mouse arthritis [36] and RA [37].

Briefly, normal human synovium harbors MerTK^+^/CD206^+^ lining MP with a protective/repair alternative M2 phenotype. They include a TREM2^+^ subset (CX3 chemokine receptor 1 (CX3CR1)^+^ counterpart in mice) and a folate receptor β (FOLRβ)^+^ subset (CX3CR1^+^ and RELM-α counterparts in mice) [36,37]. On the contrary, rheumatoid synovium harbors MerTK^-^ CD206^-^ sub-lining MP with a pro-inflammatory, classical M1 phenotype. They include a HLA^+^ subset (CCR2/Arg1^+^ and CCR2/IL1B^+^ counterparts in mice) and a CD48^+^ subset (CCR2/Arg1^+^ counterpart in mice) [36,37].

#### 2.1.2. Ontogeny of Synovial MP

As many other tissues, synovium contains tissue-resident and blood monocyte derived MP. Traditionally, synovial MP were thought to derive from fetal and adult hematopoiesis [38]. Until recent advances with high-throughput technologies, little was known about the relative contribution of these two lineages to the development and function of joints in healthy condition and arthritis. This approach combined with relevant mouse models of arthritis, allowed better deciphering of this conundrum.

Mouse models suggest that tissue-resident MP from the lining layer originate from embryonic hematopoiesis [36]. A recent study associating fate-mapping strategy with 3D-microscopy and single cell RNA sequencing (scRNA-seq) showed that CX3CR1^+^ tissue-resident MP of synovium replenish from the proliferation of CX3CR1^-^ MHC class II^+^ sub-lining mononuclear cells [36].

During arthritis, in addition to these tissue-resident MP, the subset of infiltrating synovial MP derives from blood monocytes and actively participate to arthritis [36,39,40].

#### 2.1.3. Roles of MP in the Synovium

MP are normal components of the healthy synovium. As such, they contribute to the maintenance of normal homeostasis of the joint, especially as immune sentinels. Firstly, phagocytosis is a hallmark of MP [1]. This process enables tissue-resident MP to clear both the synovial tissue and the synovial fluid from pro-inflammatory particles and apoptotic cells. Of note, MP expressing tyrosine kinase receptors from the Tyro/Axl/MerTK (TAM) family are responsible for the efferocytosis, i.e., phagocytosis of apoptotic cells [41,42], and subsequently drive pro-resolving effects [37,43]. Secondly, tissue-resident MP secrete anti-inflammatory mediators, such as IL10, IL1-ra, and osteoprotegerin [44]. These mediators reduce two major components of a destructive arthritis, namely, local inflammation and bone erosion. Thirdly, recent scRNA-seq data from fluorescent activated cell sorted tissue-resident MP in rheumatoid synovium, showed that the following subsets, MerTK^pos^TREM2^high^ and MerTK^pos^LYVE1^pos^, displayed a transcriptomic profile of negative regulators of inflammation. In addition, in vitro functional tests confirmed their ability to generate pro-resolving lipids and to induce a repair response by FLS [37]. Interestingly, recent histological data highlighted a synovial dynamic membrane-like structure, with an unexpected role of the CX3CR1^+^ MP subset from the lining layer [36]. These barrier-forming MP express tight junction proteins (JAM1, ZO-1, claudin 1), potentially preventing the invasion of the joint by infiltrating inflammatory immune cells. However, how these protective properties are affected by arthritis’ course needs clarification, especially since data from the literature suggest that some anti-inflammatory mechanisms are defective in RA [45].

On the contrary, histological studies have shown that the number of CD68^+^ infiltrating MP in the sub-lining layer is correlated with disease activity [46,47]. This parameter is also a good biomarker of therapeutic response [48], as the variation of the number of MP in the sub-lining layer accounts for 76% of the variation of disease activity under treatment [46]. The abundance of the infiltration of MP is positively correlated with structural severity [49].

These histological findings are in line with the major contribution of the inflammatory MP to arthritis’ course, as observed in rodent models. For instance, in adjuvant-induced arthritis rat model, the specific targeting of both inflammatory MP and osteoclasts by celastrol (an apoptosis inducer) is associated with inflammatory remission and bone erosion repair [50]. At the cellular and molecular levels, the pro-inflammatory role of synovial MP is well described. They are major producers of inflammatory cytokines: TNF, IL1, and IL6 [51]. Interestingly, RA patients present higher levels of these inflammatory cytokines than osteo-arthritic controls. The central role of these cytokines explains their therapeutic targeting in RA (TNF blockers: etanercept, adalimimab, infliximab, certolizumab, golimumab; IL6 blocker: tocilizumab; IL1 blocker: anakinra). Infiltrating MP also produce ROS and NO, perpetuating local inflammation [52]. They facilitate neo-angiogenesis via their production of vascular endothelial growth factor (VEGF), fibroblast growth factor (FGF) β, platelet-derived growth factor (PDGF), TNF, and IL6 [53]. Their interaction with FLS through inflammatory cytokines leads to cartilage breakdown by matrix metalloproteinases (MMP) 1, 3, and 13 [54]. A subset of inflammatory MP referred to as heparin-binding EGF-like growth factor (HBEGF)^+^ MP is enriched in RA tissues [55]. These are activated by FLS and TNF, and subsequently produce inflammatory soluble mediators such as IL1 and the EGF growth factors, and epiregulin, therefore favoring FLS invasiveness. Moreover, MP can represent a precursor for osteoclasts. The osteoclast is a specialized MP, resulting from the fusion of myelomonocytic precursors, and responsible for bone erosion [56]. Its ontogeny in humans is still matter of debate, but accumulating evidences showed that it may derive from an embryonic precursor in homeostatic condition (as tissue-resident MP), and alternatively derive from blood monocytes or transdifferentiate from MP, especially in the context of arthritis [57,58]. Finally, inflammatory MP also interact with other immune cells, contributing to their activation through antigen presentation with MHC class II molecules, T cell co-stimulation with CD80/86, and cytokines production. Indeed, this crosstalk is bi-directional and participates in MP activation as well.

#### 2.1.4. (Re)programming Synovial MP

Programming/reprogramming of MP refers to the purpose of shaping the response of these cells to switch them to their steady state functioning. Indeed, MP can adapt to their microenvironment and activate accordingly to respond to the immunological challenge. During RA, local stimuli present in both the synovial tissue and the synovial fluid are able to activate MP. Receptors of innate immunity, such as TLR2 and 4, are involved in RA pathogeny [59]. Many authors consider that pathogens from mucosal sites can trigger rheumatoid auto-immunity. *Aggregatibacter actinomycetemcomitans* and *Porphyromonas gingivalis*, two oral pathogens involved in parodontopathy, are able to generate citrullinated peptides [60], especially from fibrinogen and alpha-enolase, thus increasing the burden of auto-antigens targeted by rheumatoid autoantibodies, namely, anti-citrullinated peptides antibodies (ACPA). Interestingly, both pathogens are recognized by TLR4, or TLR2 for the latter [61,62]. In addition, TLR4 on MP is also able to recognize endogenous ligands relevant in RA pathogeny: native articular proteins, i.e., fibrinogen, collagen [63], and citrullinated peptides [64].

Although generating citrullinated peptides is not pathogenic, the development of an autoimmunity against these ACPA is the specific pathogenic process of RA. Another mode of MP activation in RA involves immunoglobulin binding through Fc gamma receptors (FcγR). Binding of ACPA-containing immune complexes by CD32a (FcγRII) induces a strong TNF production, especially when comparing monocyte-derived MP from RA patients to healthy controls [65]. This ACPA-dependent inflammatory response generates IL1 and IL6 alongside TNF, and is amplified by the presence of the other rheumatoid auto-antibody: rheumatoid factor (RF) [66]. Interestingly, MP polarized by M-CSF seem more potent producers of inflammatory mediators in the presence of immune complexes containing ACPA [67].

Cytokines are classical activators of MP. In RA, cytokine production is provided by MP themselves and other infiltrating immune cells (especially T lymphocytes), and FLS [38,68]. First attempts to characterize MP polarization used IFNγ, IL4, IL10, M-CSF, and GM-CSF to define prototypical phenotypes. We currently know that this view with discrete phenotypes, although useful to describe the polarization process, does not thoroughly describe the complexity of the phenomenon in pathology. For example, M-CSF, described as a cytokine polarizing toward a non-inflammatory phenotype, promotes proliferation, survival, maturation, and activation of monocytes and MP in arthritis [67,69,70]. Moreover, M-CSF is required alongside RANKL to generate osteoclast, the effector of bone resorption in arthritis [71]. GM-CSF is largely produced in RA by sub-lining CD90^+^ FAP^+^ synovial fibroblasts, CD90^+^ activated endothelium, and CD163^+^ MP and represents another activating cytokine for MP [72]. This role is confirmed by the positive results of its therapeutic targeting by mavrilimumab in RA [73]. IFNs are identified as major drivers of RA pathogeny. The increased expression of type I IFN and STAT1 in RA is also known as “IFN signature” [74]. Interestingly, blood IFN signature associates with infiltration of synovial B and plasma cells [35]. Recent data showed that the disease signature also includes IFNγ-mediated repression [75]. IFNγ represses M2-like gene expression associated with homeostatic, repair, and anti-inflammatory functions. This involves the inactivation and disassembly of enhancers associated with M2 genes, by targeting MAF, a M2-gene regulator. Importantly, RA synovial MP showed lower amounts of MAF mRNA, higher amounts of STAT1 and IRF1 mRNA, and higher “repression profile” (“Negative IFNγ signature”) than do synovial MP from healthy donors [75].

Another aspect of MP activation is the impact of microRNAs (miRs). Evidences accumulate conferring relevance in RA. MiR-155 is the prototypical miR responsible for pro-inflammatory regulation in monocytes and MP [76,77,78] and associated with disease activity. Many other miRs (miR-16, miR-let7a, miR-33, miR-125a, miR-223) have been described in the context of RA. Some, like miR-146a, provide a potential anti-inflammatory modulation [79].

To a certain extent, MP are also able to re-adapt to changes in their microenvironment. This plasticity allows a reprogramming of MP. Such a phenomenon has been described in arthritis, using the crosstalk of MP with other actors involved in arthritis. In ex vivo experiments from collagen antibody induced arthritis (CAIA) mouse model, factors secreted by arthritis tissue-derived synovial fibroblasts enhanced glycolytic and oxidative metabolism, and subsequently lifespan and inflammatory response in arthritis tissue-derived synovial MP [80]. One may hypothesize that primary dysfunction in FLS during RA induces reprogramming of healthy MP towards an inflammatory phenotype. MP metabolic reprogramming could represent a future therapeutic target. This is exemplified by the selective inhibition of branched-chain aminotransferase 1 (BCAT1) resulting in decreased oxygen consumption and glycolysis in human primary MP, and consequently in the reduction of pro-inflammatory signature in the CAIA mouse model [81].

Adipose tissue-derived stem cells (ADSC) have also been used as source for MP reprogramming. In ex vivo experiments using RA synovial fluid to condition ADSC, the synovial pro-inflammatory environment triggered immunomodulatory potential in ADSC in a TNF/NF-κB-dependent manner. This immunomodulatory potential of ADSC was characterized by the expression of cyclooxygenase-2 (COX2), indoleamine-1,2-dioxygenase (IDO), IL6, TNF stimulated gene 6 (TSG6), intercellular adhesion molecule 1 (ICAM-1), vascular cell adhesion molecule 1 (VCAM-1), and programmed death-ligand 1 (PD-L1). In a co-culture setting, conditioned ADSC modulate the phenotype of the inflammatory MP, especially by reducing their expression of CD40 and CD80. This kind of approach may be beneficial in the future to treat arthritis.

### 2.2. Therapeutic Modulation of Macrophages in Arthritis

#### 2.2.1. Impact of RA Treatment on Synovitis and MP

Data regarding the impact of disease modifying anti rheumatic drugs (DMARDs) on synovial MP at the molecular level are scarce. Most studies focus on in vitro/ex vivo models based on monocyte-derived MP. This topic has been extensively reviewed in [82]; the following section will be dedicated to human findings with RA samples.

Conventional synthetic (cs)DMARDs are the first line treatment in RA. This family includes methotrexate (MTX; a folate antagonist), the cornerstone of RA treatment, leflunomide (LFN; an inhibitor of pyrimidine synthesis), and sulfasalazine (SFZ; an anti-bacterial and anti-inflammatory agent). All these drugs can reduce infiltration of MP, the importance of the reduction in sub-lining MP correlating with the clinical response [46]. At the molecular level, most data come from MTX studies and reveal an inhibition of pro-inflammatory mediators (IL1 and IL6) produced by MP, and the induction of a tolerant state dependent on NF-κB suppressor A20 (TNFAIP3) [83]. MTX also modulates plasma metabolites, such as itaconate [84]. This observation suggests anti-inflammatory effects mediated through the anti-inflammatory and anti-oxidative transcription factor NRF2 (nuclear respiratory factor). Like MTX, SZP and LFN reduce the production of central pro-inflammatory cytokine in RA: TNF, IL1, and IL6 [85].

Biological (b)DMARDs are recommended for patients with moderate to severe RA and inadequate response to a first line csDMARD. TNF blockers (TNFi) are the most widely used. TNFi reduce the sub-lining infiltration by CD68^+^ MP [46,86]. At the molecular level, TNFi demonstrated a specific modulation of MP phenotype, favoring a phagocytic phenotype expressing higher membrane CD163 and MerTK, and reducing membrane (CD40, CD80) and secreted (TNF, IL6, IL12) pro-inflammatory parameters. This negative regulation of inflammation involved an IL10 production induced by suppressor of cytokine signaling 3 (SOCS3) and GAS6 in a STAT3-dependent manner [87]. Another less known mechanism of action of TNFi is the induction of a “reverse signalling”. Reverse signalling is the induction of an anti-inflammatory response initiated by the transmembrane TNF (precursor form of TNF on TNF-producing cells), and triggered by its binding, in particular to TNFi [88]. This impact of TNFi has been described in monocytes and MP and is likely to explain a part of the clinical response in RA [89].

Tocilizumab (TCZ) is an anti-IL6R. Results regarding the reduction of CD68^+^ infiltration are variable [90,91]. TCZ reduces the production of pro-inflammatory mediators. However, described impact on MP phenotype is variable as well, potentially depending on the experimental model. Some reports highlighted the induction of an anti-inflammatory phenotype driven by peroxisome proliferator activated receptor γ (PPARγ). Overall, it seems that TCZ impact is more pronounced on CD3^+^ T cells than on MP.

Abatacept (ABA) is a fusion protein (extracellular domain of cytotoxic-T-lymphocyte antigen 4 (CTLA-4) protein and Fc region of an IgG1) designed to disrupt the interaction between T cells and antigen presenting cells. When assessed in synovium of treated patients, it does not seem to reduce immune infiltration but this observation has to be interpreted considering synovia pathotypes [92]. ABA indirectly downregulates the activation of MP by T cells and subsequently IL6, TNF, IL1β, IL12p70, and TGFβ by RA synovial MP and monocyte-derived MP in co-culture experiments [93,94]. It also blocks the autoantibody-driven (ACPA, RF) inflammatory activation of monocyte-derived MP in an IDO-dependent manner [95].

Rituximab (RTX) is an anti-CD20 targeting B cells. No clear reduction in the infiltration of MP has been reported [96]. However, despite not directly targeting MP, in treated patients, RTX alters monocyte-derived MP functions with an increase in B cell activating factor (BAFF), IL10, and CD86 expression, as well as a decrease in TNF secretion [97]. It should be noted that RTX, ABA, TCZ, and adalimumab (a TNFi) share a common transcriptomic downregulation of genes involved in both T cell and myeloid leukocyte activation pathways, such as LCK, STAT1, STAT3, JAK2, and JAK3, irrespective of the primary drug’s target [92].

The most recent therapeutic class developed to treat RA is the one of JAK inhibitors (JAKi). Indeed, the pathogeny of RA involves several JAK-STAT-dependent cytokines, such as IL2, IL6, GM-CSF, and IFNα/β. The four JAKi currently approved in RA: tofacitinib, baricitinib, upadacitinib and filgotinib, have different JAK specificities, but share a common targeting for JAK1 [98,99]. Importantly, the so-called specificity of these inhibitors is not absolute, and depends on the dose, the cell type used for assessment, and the experimental model [98,99,100]. To date, data about the impact of JAKi on human RA synovium are scarce. In one study, the use of tofacitinib in RA patients with inadequate response to MTX did not reduce the synovial immune infiltration (including CD68^+^ sub-lining MP) at day 28 compared to baseline [101]. However, the drug significantly reduced the mRNA expression of synovial chemokines (CXCL13, CXCL10/IP-10, CCL2/MCP-1), and MMP (MMP-1, MMP-3), with a correlation between the synovial levels of pSTAT1 and pSTAT3 and the clinical response at month four. At the cellular level, JAKi modulate the activation of MP. In vitro studies confirmed the inhibition of pro-inflammatory target genes, including CXC chemokines and IFN/STAT1 signature in monocyte-derived MP from RA synovial fluid [102]. Unexpectedly, some papers reported a reinforcement of M1 phenotype with LPS stimulation [103]. It is highly probable that models using LPS stimulation on isolated and monocyte-derived MP, highlight the inhibition of the IL10/STAT3 negative feedback on inflammation by JAKi. Additionally, JAKi are thought to indirectly affect MP activation processes while targeting other actors of rheumatoid synovitis (synovial fibroblasts, T lymphocytes, etc.) and reducing their ability to provide pro-inflammatory triggers for the MP [104,105].

#### 2.2.2. Theranostics and MP in Arthritis

Synovial MP heterogeneity may impact RA outcome and therapeutic response. Recent studies using histology and RNA-seq approach assessed MP-derived parameters for this perspective.

In patients from the adalimumab actemra (ADACTA) trial treated with TNFi adalimumab), baseline synovial myeloid gene signature expression was higher in patients with good, compared with poor European league against rheumatism (EULAR), clinical response at week 16 [106]. Conversely, a pauci-immune synovial pathotype predicts inadequate response to TNFi (certolizumab pegol) [107]. These findings are consistent with the preferential impact of TNFi on myeloid cells, especially MP.

In the Pathobiology of Early Arthritis Cohort (PEAC), patients with a baseline lympho-myeloid pathotype (versus diffuse-myeloid or pauci-immune) require bDMARD significantly more often at 1 year [108]. This emphasizes the importance of lymphocyte/MP crosstalk in RA pathogeny. Synovium gene modules related to chemokines/inflammatory molecules in myeloid cells and to TLR and inflammatory signalling were correlated with disease activity (DAS28-CRP) variation from baseline to month 6 [35]. The analysis of differential expression of synovial single-cell-annotated WGCNA (weighted gene correlation network analysis) modules between EULAR DAS28-CRP responders (good and moderate) and non-responders showed higher expression of TLR signalling related genes in responders [35].

In addition, a low ratio (≤2.5) of MerTK^+^ CD206^+^/MerTK^-^ CD206^-^ and a low proportion (≤47.5%) of MerTK^+^ CD206^+^ synovial MP at the timepoint of treatment tapering/discontinuation was independently associated with non-persistent remission, i.e., disease flare [37]. Usefulness of these parameters to predict sustained remission needs confirmation. 

Overall, considering parameters derived from synovial MP seems to be relevant for theranostic perspectives. To date however, using MP alone does not appear sufficiently efficient, since T and B cells are also required to fully reflect disease course. This type of approach needs refining in order to be used at the individual scale for personalized medicine perspective.

#### 2.2.3. Reprogramming MP with Poly(phosphorhydrazone) (PPH) Dendrimers

Dendrimers are defined as hyperbranched and multifunctional “tree-like” molecules. Unlike linear polymers, they are not synthesized by polymerization reactions but rather by step-by-step methods thanks to the iteration of sequential reactions. At the end of each sequence, a supplemental series of branches is added (so-called “generation”) encompassing twice or three times the number of branches of the previous generation. The first series of branches is linked to the core of the dendrimer. The multiplication of the number of branches from one generation to the next is enabled by the points of divergence grafted at the end of each branch (Figure 4A). This process leads to arborescent molecules that are ended by surface functions affording the desired properties.

From the very beginning, dendrimers have been very attractive for biological and medical applications. The main reason for that is their multivalency that enables particular behavior towards biological materials [109]. Their advent was also concomitant with the booming of nanotechnologies in the 1980s [110]. Amongst the numerous families of dendrimers, phosphorus-based dendrimers are built on a cyclotriphosphazene core on which PPH branches are added. Among these PPH dendrimers, the first-generation dendrimer (i.e., having a single series of branches) ended by twelve azabisphosphonate groups, the so-called dendrimer ABP (Figure 4B), has shown immunomodulatory properties towards different human primary immune cells. It inhibits the proliferation of CD4^+^ T lymphocytes [111], it also inhibits the maturation of monocyte-derived DC [112], and it activates monocytes and MP towards an anti-inflammatory phenotype [113]. We have shown that a consequence of the latter property is the ex vivo amplification of natural killer (NK) cells [114]. Based on these results, we have challenged the immunomodulatory and anti-inflammatory properties of the dendrimer ABP in several rodent models of acute (endotoxin-induced uveitis [115]), and chronic (arthritis [69,116], multiple sclerosis [117], and psoriasis [118]) inflammatory disorders. In mouse models of experimental arthritis (the IL1-ra KO and the K/BxN models), we have shown that the dendrimer ABP efficiently controls the development of the diseases. Indeed, by targeting the pro-inflammatory monocytes and MP and reversing them towards an anti-inflammatory metabolism, the dendrimer ABP controls the three main pathophysiological features of arthritis: systemic and joint inflammation (by decreasing the secretion of pro-inflammatory cytokines), cartilage degradation (by decreasing the secretion of MMP), and bone resorption (by inhibiting both the differentiation of monocytes and the transdifferentiation of DC in osteoclasts) [69]. We have shown that the same effects can be described in human rheumatoid synovial tissue coming from arthroplasty [69]. Overall, we have shown in all these in vivo assays that the immunomodulatory effects of the dendrimer ABP contributes to the production of IL10, the paradigm of anti-inflammatory cytokines [119]. Some other dendrimers also have anti-inflammatory properties, but they do not target inflammatory monocytes and MP to reprogram them and promote the resolution of inflammation [120]. Favorable tolerance, safety, and biodistribution preliminary results [121,122], dendrimer ABP is on the way towards regulatory preclinical studies for the treatment of RA.

## 3. Managing Tumor-Associated Macrophages (TAM)

### 3.1. Origin and Functions of TAM Phagocytosis Receptors

Tumors are now considered as complex systems composed of tumor cells surrounded by many other different cell types constituting the TME. Among these cells, MP represent a major part of the immune cells within the TME and are called tumor associated MP (TAM). Until recently, TAM were considered to exclusively originate from blood-derived MP infiltrating the tumor tissue before undergoing differentiation. However, it is clear today that tissue-resident MP can coexist in the tumor with infiltrated monocytes and MP [123]. Monocytes/MP are attracted from blood, bone marrow, and spleen to the tumor site thanks to CCL2, CCL5, and CSF-1 produced by the tumors cells, fibroblasts, endothelial cells, and even by the TAM themselves [124]. CSF-1 is also highly involved in the MP survival and polarization in the tumors [125]. Depending on the molecular and cellular components in the TME and the tumor stage, TAM can display either a M1 or M2 phenotype (Figure 5A) [126]. In early-stage tumor development, IFNα polarizes resident MP towards an M1 phenotype and activates the infiltration of blood derived-M1 MP. M1 TAM are therefore able to phagocytize tumor cells and to release pro-inflammatory factors for recruitment and activation of other immune effectors cells [127,128,129,130]. However, in advanced tumors, anti-inflammatory mediators released in the TME, such as CSF-1, CCL2, IL13, IL4, IL10, and prostaglandin E2 (PGE2), can revert the anti-tumor program and favor a switch of TAM into an M2 phenotype with pro-tumor and immunosuppressive functions [131,132]. Indeed, TAM are able to promote tumor cell genetic instability, proliferation, migration, and resistance to apoptosis, to immune attacks, and to therapies [133]. Some interleukins, such as IL23, IL17, and IL6, but also iron, produced by TAM activate cancer cell proliferation [134,135,136]. Tumor cell migration and invasion are promoted by IL6 and also iron, and by MMP or CCL18 secreted by TAM [137,138]. TAM are also able to produce some pro-angiogenic factors, such as TGFβ, VEGF, or PDGF, to induce angiogenesis and lymphangiogenesis [139]. The immunosuppressive functions of TAM are implemented through the secretion of IL10, TGFβ, or PGE2 that neutralize recruitment and function of cytotoxic CD8 T cells and NK cells, and induce Treg functions [140,141,142]. Furthermore, TAM protect tumor cells from therapies when their depletion significantly increases the response to chemo or radiotherapies in breast cancer mouse models for instance [143,144].

TAM are therefore good candidates to target in TME, and several therapy strategies modulating TAM functions, infiltration, and number are emerging.

### 3.2. Reprogramming TAM

Regarding the plasticity of MP populations, TAM with pro-tumor functions can also be reprogrammed toward a tumoricidal phenotype to restore their anti-tumor properties. This reprogramming can be achieved using different tools targeting either surface markers or signaling molecules, or modifying the metabolism: (i) specific antibodies; (ii) small molecules; (iii) functionalized nanoparticles; and (iv) other systems, such as hydrogels, liposomes, exosomes, and nanocrystals (Figure 5B).

#### 3.2.1. Specific Antibodies and Peptides

Different markers at the surface of TAM can be targeted to switch their phenotype, such as CD206, signal regulatory protein alpha (SIRPα), CD40, CCR5, MARCO, or CSF1-R.

RP-182 is a synthetic 10-mer amphipathic analog that selectively induces a conformational switch of the mannose receptor CD206 leading to a reprogramming of TAM in a M1-like phenotype with a recovery of endocytosis, phagosome-lysosome formation, and autophagy functions. This peptide has been shown to participate in the suppression of tumor growth in syngeneic and autochthonous mouse cancer models [145].

SIRPα expressed by TAM interacts with CD47 at the surface of cancer cells, and transmits the “don’t eat me” signal. Engineered SIRPα-Fc fusion is able to restore MP’s ability to phagocytize cancer cells and prime cytotoxic CD8 T cells. One of these fusion proteins, namely, TTI-621, is currently being tested in phase I clinical trials in AML and other hematological malignancies, pediatric brain tumors, and some multiple solid tumors (trial numbers NCT02678338 and NCT03957096). To minimize the off-target toxicity (transient anemia), bispecific antibodies targeting both CD47 and tumor-associated antigens were recently developed with promising results [146].

CD40 is also an important marker of TAM, as its ligation with the CD40 ligand at the T cell surface stimulates T cell–based anti-tumor responses. Modified anti-CD40 antibody, presenting five point mutations in the Fc domain (CP-870,893), was shown to increase pancreatic carcinoma sensitivity to chemotherapy in a pancreatic ductal adenocarcinoma (PDAC) mouse model associated with a switch of infiltrated MP due to an increase in CCL2 expression and IFNγ in these mice [147]. This last study opened promising perspectives for combination therapies. CCR5 antagonists such as maraviroc provided an anti-tumor effect in a phase I trial in patients with liver metastasis of advanced refractory colorectal cancer correlated with a MP repolarization [148].

An anti-MARCO mAb was developed and has been shown as having an anti-tumor activity in breast and colon carcinoma, and in melanoma models through reprogramming of TAM populations to a pro-inflammatory phenotype and increasing tumor immunogenicity. This was shown as being dependent on the inhibitory Fc-receptor, FcγRIIB [149].

Many studies demonstrated that combining chemotherapy or immunotherapy with the blockade of CSF1-R could improve anti-tumor T cell responses. For instance, in a mouse model of PDAC, the use of an anti-CSF1-R antibody in combination with antagonists of PD-1 and CTLA-4 enhanced antigen presentation and productive anti-tumor T cell responses leading to tumor regressions [150].

#### 3.2.2. Small Molecules

The pro-tumor functions of TAM are achieved through some specific signaling pathways and a metabolism based on the Warburg’s effect that tends to favor a specialized lactic fermentation over an aerobic respiration pathway in mitochondria [151,152]. Lactate is secreted in the extracellular environment and can be re-used by cells of the tumor themselves. Targeting molecules of these pathways and modifying metabolism thanks to small molecules is also a way to reprogram TAM.

Some synthetic TLR ligands were tested in cancer, such as the TLR3 agonist poly(I:C), a synthetic double stranded-RNA, which activates the NF-κB pathway, leading to pro-inflammatory M1 polarization with production of type I IFNγ [153]. Clinical trials in metastatic cancers with imiquimod, a synthetic imidazoquinoline which activates TLR7, showed histological tumor regression and an increase in lymphoid immune infiltration [154]. Intratumoral delivery of cytosine–phosphate–guanine oligodeoxynucleotides (CpG ODN), a TLR9 agonist, or of R848, a TLR7/8 agonist, showed tumoricidal activity in melanoma and breast cancer mouse models [155,156]. Bacterial ligands, such as the attenuated ΔactA/ΔinlB strain of *Listeria monocytogenes* can also stimulate TLRs. When introduced into the aggressive ID8-Defb29/Vegf-A mouse ovarian carcinoma, TAM were activated to phagocytize cancer cells and became more immuno-stimulating [157].

The FOLRβ being upregulated in TAM, treatment of mice with orthotopic breast cancer with a folate-targeted TLR7 agonist showed a decrease of tumor mass and reprogramming of TME including TAM [158].

TLRs signaling involves many kinases which can be inhibited by small molecules. Inhibitors of ubiquitinase enzymes (USP) were shown to mediate TAM reprogramming by activating the p38 MAPK pathway [159]. MAPK interacting protein kinase 1 (MNK1) which participates in mediating high-fat-diet-induced insulin resistance can be inhibited by cercosporamide, an anti-fungal natural product, leading to the inhibition of the phosphorylation of eIF4E, a translation initiation factor, and to the reprogramming of TAM toward a pro-inflammatory phenotype [160]. Other protein kinases, such as Ca^2+^/calmodulin-dependent protein kinase II (CAMKII), which regulates calcium signaling in health and disease, was specifically inhibited by the small molecule KN93, a metoxybenzenesulphonamide, leading to the decrease of M2-associated markers, such as CD206 or Arg1, and an increase of M1-related molecules, such as CD86 or iNos2 [161]. Receptor interacting protein (RIP) kinases, a crucial regulators of cell survival and death, were specifically inhibited by the molecule GSK’963, leading to reprogramming of TAM toward an MHCII^hi^ TNFα^+^ IFNγ^+^ immunogenic phenotype in a STAT1-dependent manner [162]. STAT1 signaling as and NF-κB were also shown as activated by polysaccharides isolated from the root of *Ilex asprella* (IAPS-2) or from Lachnum (a sort of mushroom) promoting secretion of anti-tumor cytokines by TAM and increasing animal survival rate in a sarcoma mouse model [163,164]. In the same way, STAT6, also a major signal transducer activated by IL13, can be inhibited by different synthetic molecules (AS1517499, TMC-264, A771726), leading to an inhibition of tumor growth in the 4T1 mammary tumor model and to a modification of genetic markers for TAM infiltration [165]. STAT3 was shown to mediate the effect of rapamycin, a cyclic molecule used as immunosuppressor in cases of grafts, through inhibition of mTOR, promoting MP-mediated antitumor effect in a hepatocarcinoma mouse model [166]. Moreover, inhibitors of PI3kγ, a molecule upstream of mTOR, are able to promote TAM-immunostimulatory responses in several cancer models [167]. Beside, Bruton’s tyrosine kinase (BTK) inhibition by ibrutinib reprograms MP toward an M1-like phenotype that promotes CD8 T cells cytotoxicity and suppresses tumor growth in PDAC [168].

M1 genes, such as iNOS and CXCL9, can also be upregulated concomitantly with the downregulated expression of M2 genes, Arg1 and CD206, following the inhibition of the CYP4X1 monooxygenase or the activation of the autophagy-induced RelB/p52 by flavonoids in glioma or hepatocellular carcinoma models [169,170].

Chloroquine, which was recently shown as a promising antitumor agent, is finally a potent immune modulator mediating its antitumor efficacy via the reprogramming of TAM from M2 to M1-like phenotype [171]. This molecule increases MP lysosomal pH, causing the release of calcium, inducing p38 and NF-κB activation, thus polarizing TAM toward an M1 phenotype.

The regulation of the MP mitochondrial function can also lead to the reprogramming of TAM. Indeed, downregulation of the gene encoding pyruvate dehydrogenase is avoided by dampening the NRF1 degradation under hypoxia, which minimizes the Warburg’s effect and promotes M1 polarization of TAM [172]. Cancer cells are known to secrete large amounts of lactate through glycolysis, which is recognized by Gpr132 at the surface of TAM promoting an M2 polarization. PPARγ agonists, which are suppressive for the Gpr132 axis, were successfully used to desensitize TAM to lactate stimulation in a breast tumor model [173,174]. The administration of type 2 diabetes drug metformin to an osteosarcoma mouse model was also shown to inhibit tumor growth associated with an increase of production of IL12 and TNF by TAM, and an elevation of MHC class II and a reduction of CD206 expression by TAM [175]. The metformin, decreasing the Oxphos status and increasing glycolysis, can therefore participate in the shift from M2- to M1-like phenotype of MP in the tumor.

#### 3.2.3. Functionalized Nanoparticles

Some depolarizing agents that are too toxic to be delivered as is, can be packaged in structures lowering their toxicity and increasing their efficiency.

Iron oxide nanoparticles (NP) loaded with L-arginine and sealed with poly(acrylic acid) (PAA) can release L-arginine based on pH-responsive PAA and produce nitric oxide (NO) via iNOS overexpressed in TAM. In vitro and in vivo studies showed that these NP could reprogram M2 to M1 MP producing high levels of NO and TNF, leading to synergistic tumor therapy [176].

Co-encapsulated photosensitizers indocyanine green (ICG) and titanium dioxide (TiO_2_) with amsmonium bicarbonate (NH_4_HCO_3_) in mannose-modified PEGylated poly(lactic-co-glycolic acid) (PLGA) nanoparticles were synthetized for the delivery of photosensitizers to endosome/lysosome or cytoplasm of TAM [177]. Upon internalization of these NP by TAM through mannose receptor-mediated endocytosis, NH_4_HCO_3_ produces CO_2_ and NH_3_ to disrupt the endosome/lysosome membrane thus releasing photosensitizers into the cytoplasm. The laser illumination then induces generation of ROS inside TAM inducing phenotype switching of M2 to M1 [177].

Hyaluronic acid-coated mannan-conjugated MnO_2_ particles (Man-HA-MnO_2_) treatment of a breast tumor mouse model has been shown to significantly increase tumor oxygenation and downregulation of HIF-1α and VEGF in the tumor [178]. This was achieved thanks to the production of O_2_ and a pH regulation by the TAM after their uptake of Man-HA-MnO_2_ NP. Indeed, HA induces an M2 to M1 switching allowing production of H_2_O_2_, which can react with MnO_2_ NP to produce O_2_ and thus reduce hypoxia and modulate chemoresistance. Another kind of NP associated to MnO_2_ and HA, based on lanthanide-doped upconversion nanocrystals and combined with light-mediated photodynamic therapy (PDT), was shown as attenuating hypoxia status and synergistically reprogramming TAM in in vitro culture [179]. Mannosylated lactoferrin NP systems (Man-LF NPs) have been also developed for dual-targeting delivery of shikonin via both the mannose receptor and LRP-1. Shikonin is a liposoluble naphthoquinone pigment isolated from the traditional Chinese herb Zicao that was shown as inducing the generation of ROS and the suppression of both NF-κB-regulated gene products and pyruvate kinase. Treatment of an immunocompetent colorectal tumor mouse model with these Man-LF NPs led to a decrease of the tumor volume concomitantly with a downregulation of the M2- related markers and TGFβ expression of TAM while the expression of STAT1 and TNF was increased [180]. Other mannosylated NP systems have been used to deliver in vitro-transcribed mRNA encoding interferon regulatory factor (IRF) 5 in combination with its activating kinase IKKβ. Infusion of these NPs in mouse models of ovarian cancer, melanoma, and glioblastoma has been shown to reverse the immunosuppressive and tumor-supporting state of TAM and reprogramming them, promoting tumor regression [181].

TAM express a variety of pathologically relevant microRNAs (miR), which are small endogenous noncoding nucleic acids and which can affect the phenotype exhibited by TAM. Some miR, such as miR-125 or miR-155, are known to promote repolarization of MP from M2 toward M1 phenotype. A good way to enhance expression of these miR in TAM is to directly deliver them inside TAM through NP. HA-based NP or lipid-coated calcium phosphonate NP (CaP/miR@pMNPs) containing conjugated mannose, were successfully used as delivery systems for miR-125b and miR-155, respectively, in cancer mouse models [182,183].

#### 3.2.4. Other Systems

Systems other than NP can be loaded or associated with a specific compound to be delivered to TAM.

Corosolic acid packaged within long-circulating liposomes and coupled to an anti-CD163 antibody was shown to inhibit STAT3 activation in human M2 MP and IL10-induced gene expression thereof [184]. Liposomes can also be associated with two molecules targeting two different cells in the TME. A trastuzumab-modified, mannosylated liposome system loaded with gefitinib (kinase inhibitor) and vorinostat (histone deacetylase inhibitor) was used in a nonsmall cell lung cancer (NSCLC) mouse model for dual-targeting of both TAM and HER2-positive lung cancer cells [185]. These authors showed that treatment with these liposomes induced a modulation of the intracellular redox balance with a reprogramming of TAM toward an M1 phenotype and an increase of ROS in the cancer cells decreasing their drug-resistance.

Another delivery system was constructed from bone marrow mesenchymal stem cell (BM-MSC) exosomes, electroporation-loaded galectin (Gal)-9 siRNA, and surficially modified with oxaliplatin (OXA) prodrug as an immunogenic cell death-trigger. Gal-9 siRNA blocks the Gal-9/dectin-1 axis allowing the repolarization of TAM in M1 MP. Treatment of a pancreatic cancer mouse model with these exosomes elicited reversion of immunosuppression caused by TAM and therefore anti-tumor responses [186].

Inhibition of the CAMKII highly expressed by M2 TAM is often encountered with inefficient intracellular uptake and short circulating time. To avoid this problem, some authors incorporated CAMKII inhibitors in peptide hydrogels and succeeded in obtaining antitumor effects and MP reprogramming in mouse bearing melanoma treated with this system [161].

TAM targeting seems to be a promising way to increase efficiency of cancer therapies. The different approaches to reprogram TAM toward an antitumor phenotype listed in this review, could therefore actively support cancer therapies.

## 4. Conclusions

Over the past decades, monocytes/MP have gained a reawakening of interest from immunologists and pathologists. Both the involvement of these cells in the onset, the development and the maintenance of chronic inflammation, and their pivotal role in the TME are responsible for this return. Monocytes/MP are also key players in the resolution of inflammation when they are skewed towards anti-inflammatory activation [187]. Therefore, the repolarization of unbalanced MP, with sustained pro-inflammatory activation, is considered a relevant therapeutic strategy in chronic inflammatory diseases and cancer. Herein we have reviewed the current knowledge about the role of both inflammatory MP in rheumatoid arthritis as the paradigm of chronic inflammatory disorders, and TAM in cancer. We have also presented in detail the different therapeutic strategies that are explored and developed to repolarize these cells and offer so much hope for fighting against these diseases.

## Figures and Tables

**Figure 1 ijms-23-01496-f001:**
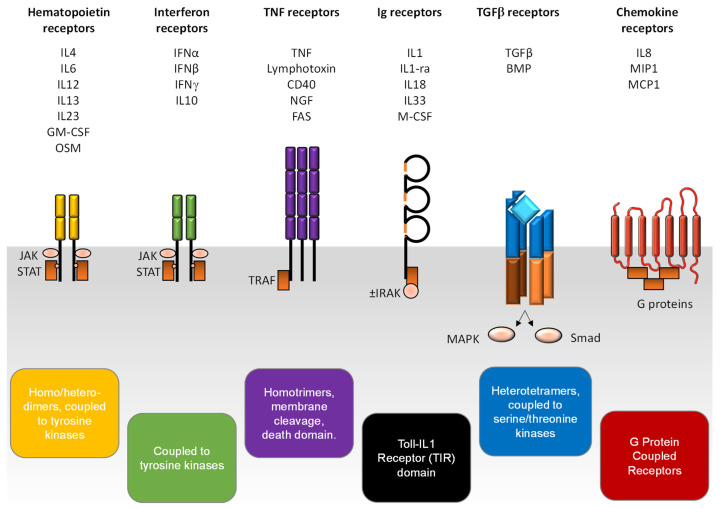
Cytokine receptors of MP. BMP: bone morphogenic protein; CD: cluster differentiation; GM-CSF: granulocyte macrophage-colony stimulating factor; IL: interleukin, IL1-ra: interleukin-1 receptor antagonist; JAK: Janus kinases; IFN: interferon; IRAK: interleukin-1 receptor-associated kinase; MAPK: mitogen-activated protein kinases; MCP1: monocyte chemoattractant protein 1; MIP1: macrophage inflammatory protein 1; NGF: nerve growth factor; OSM: oncostatin M; STAT: signal transducer and activator of transcription; TNF: tumor necrosis factor.

**Figure 2 ijms-23-01496-f002:**
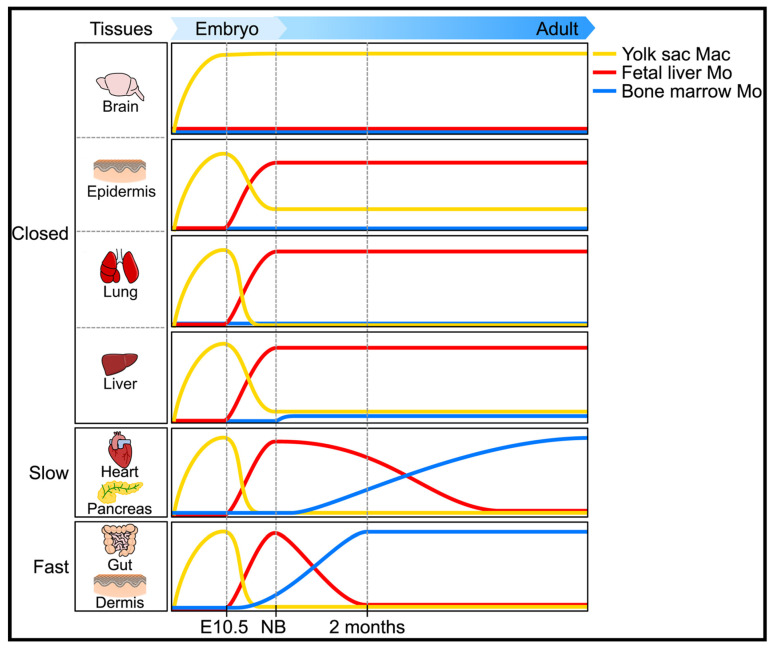
Heterogeneity of the ontogeny of resident MP from adult tissues in mouse in the steady state. Taken from [14], copyright 2016 Elsevier Inc.

**Figure 3 ijms-23-01496-f003:**
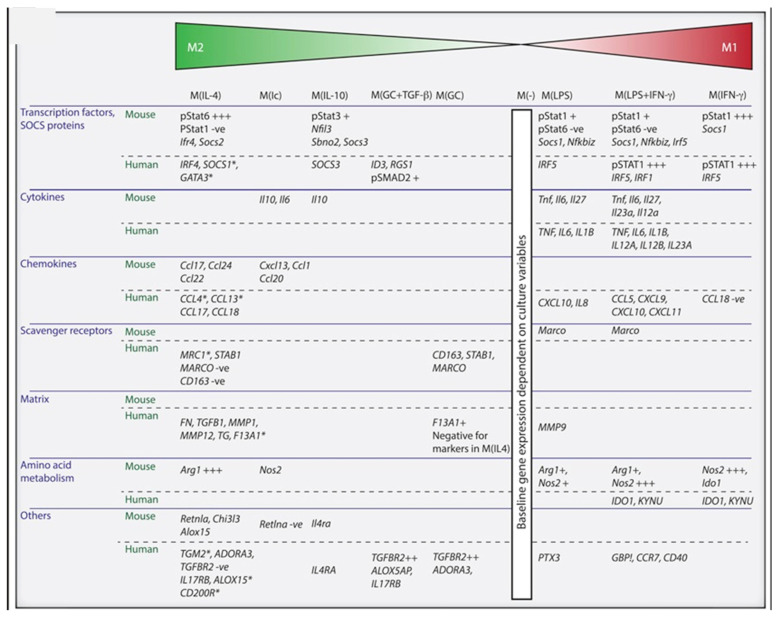
Description of activated MP. Functional classification of both mouse MP differentiated with M-CSF and monocyte-derived human MP. Taken from [32], copyright 2014 Elsevier Inc.

**Figure 4 ijms-23-01496-f004:**
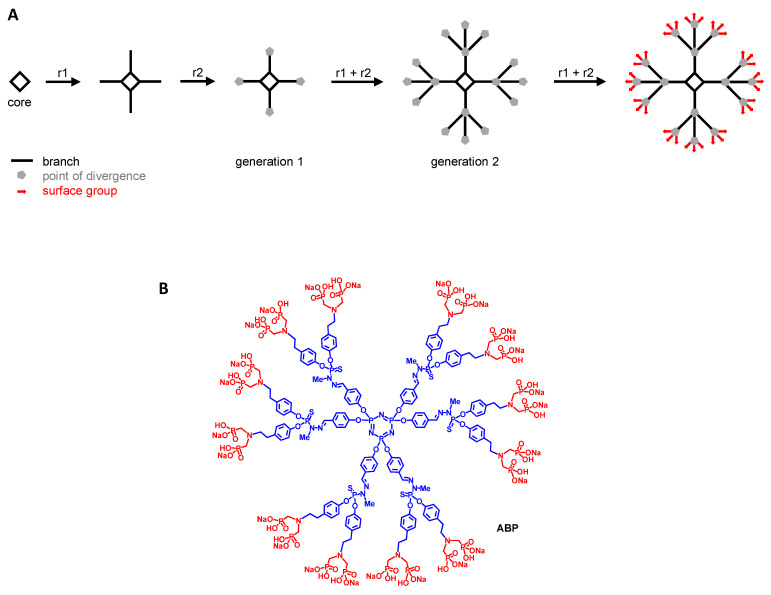
(**A**) Schematic synthesis of a generation 2 (2 series of branches) dendrimer using a tetravalent core (i.e., 4 branches in the first series) and trivalent points of divergence (i.e., 12 branches in the second series, and 36 surface groups). r1 and r2 are the reactions which are iterated to obtain the final dendrimer. (**B**) Structure of the dendrimer ABP. The cyclotriphosphazene core (N_3_P_3_) and the PPH branches (including the point of divergence) are in blue. The twelve tyramine-based (in blue) ABP surface groups are in red.

**Figure 5 ijms-23-01496-f005:**
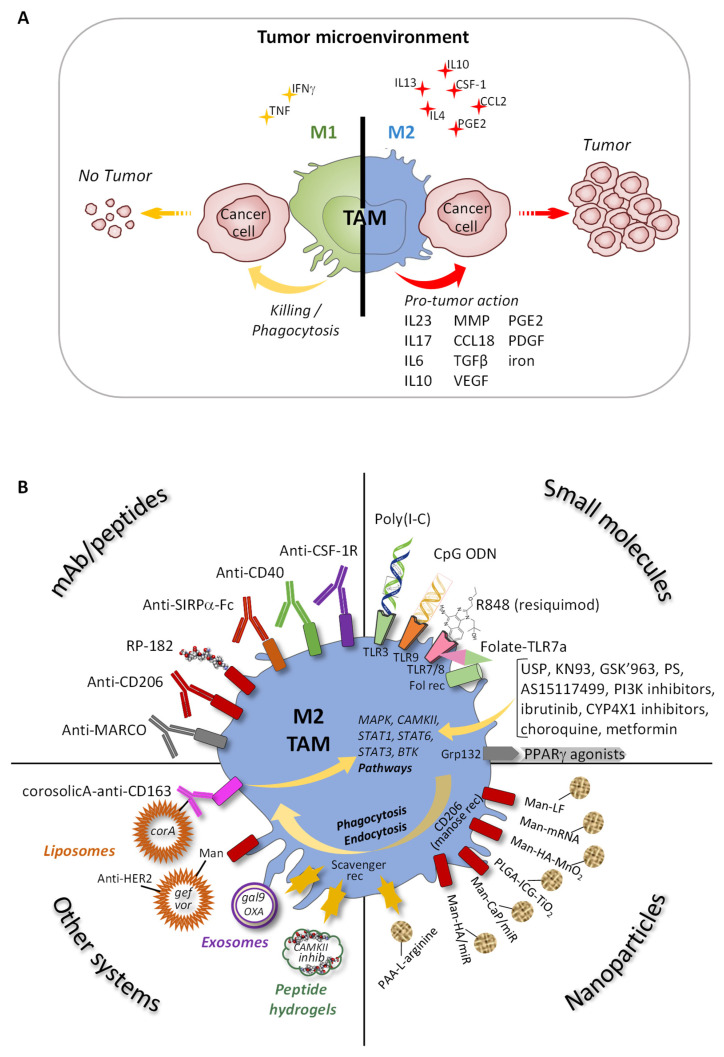
(**A**) Tumor development according to the M1 or M2 TAM phenotype in the TME: the M1 phenotype is induced by inflammatory molecules (yellow) and M2 phenotype by anti-inflammatory molecules (red). M1 TAM promote the killing of cancer cells and M2 TAM produce factors promoting tumor development. (**B**) Reprogramming of M2 TAM with: (i) monoclonal antibodies (mAb) or peptides targeting specific membranes receptors; (ii) small molecules targeting specific membrane proteins or penetrating into TAM; (iii) nanoparticles; (iv) other systems.

## Data Availability

The data that support the findings of this study are available from the corresponding author upon reasonable request.

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
