# Peer review of "Repolarization of Unbalanced Macrophages: Unmet Medical Need in Chronic Inflammation and Cancer"

_ijms, 2022, doi:10.3390/ijms23031496_

Round 1

Reviewer 1 Report

In this review, the authors discussed the roles of macrophages in chronic inflammatory disease (arthritis) and cancer, including macrophage cytokine receptors, their reprogramming, and relative strategies. The manuscript was well-organized and presented. Some minors are suggested to enhance the quality.

Abbreviations: TNF showed twice in lines 45 and 146, not consistent, such as IL6, IL-12, TGFβ and TGF-β, CX3CR1 in lines 206 first shown in line 191, miR-155 and miR155, some lacks full name such as FOLR2, etc.

Section title 2.1.4 in line 264 needs correction.

Chemical name:  NH4HCO3 (line 639) should be corrected into NH4HCO3.

Copyright of Figure 2 and Figure 3 should be acquired before accepted for publication.

Author Response

Firstly, we would like to thank the reviewer for her/his thorough reading of our manuscript and for her/his proposals to improve it. Accordingly, we have taken into account all comments and concerns. All changes are mentioned in the point-by-point answers below (italics), and are highlighted in yellow in the revised version of the manuscript.

In this review, the authors discussed the roles of macrophages in chronic inflammatory disease (arthritis) and cancer, including macrophage cytokine receptors, their reprogramming, and relative strategies. The manuscript was well-organized and presented. Some minors are suggested to enhance the quality.

Abbreviations: TNF showed twice in lines 45 and 146, not consistent, such as IL6, IL-12, TGFβ and TGF-β, CX3CR1 in lines 206 first shown in line 191, miR-155 and miR155, some lacks full name such as FOLR2, etc.

TNF showed in new line 158 has been removed (there, “TNF” is highlighted in yellow).

Inconsistent abbreviations have been homogenized throughout the text: interleukins as IL6, also TGFb, miRs as miR155.

CX3CR1 is now shown in new line 212 (highlighted in yellow) and has been removed in new line 227.

As FOLR2 is the name of the gene, it has been replaced by the name of the protein, ie, “FOLRb” (highlighted in yellow). FOLRb is shown in new line 213. Accordingly, in new line 616, the “folate receptor” has been replaced by “the FOLRb”.

Section title 2.1.4 in line 264 needs correction.

It has been corrected (highlighted in yellow).

Chemical name:  NH4HCO3 (line 639) should be corrected into NH4HCO3.

It has been corrected in new line 678 (highlighted in yellow).

Copyright of Figure 2 and Figure 3 should be acquired before accepted for publication.

Demand is ongoing.

Reviewer 2 Report

Reviewer comments and suggestions

In the current review, the authors have discussed repolarization of unbalanced MP as a therapeutic strategy to control chronic inflammation-related diseases such as Rheumatoid Arthritis, and cancer. Generally, monocytes and the tissue counterpart macrophages (MP) constitute the front line of the immune system. They can activate the immune system and monitor the immune response based on the signaling received. The balance between these dual activities is well managed and any type of unbalanced response of MP leads to pathological disorders. The authors suggested that repolarization is an important phenomenon as a therapeutic strategy.  

Decision: Minor comments

The paper was nicely written and needs some minor corrections before the final publication of the manuscript. Below are the comments for this paper to be incorporated in the revised version of the manuscript. 

  1. Section 1.3 needed to explore more on this
  2. Line 154 In my view the authors need to discuss TAM here at least 2-3 lines
  3. Line 158-59, authors discuss the vast majority of the studies but cite only one reference.
  4. Line 163-164 need to discuss figure 3
  5. Line 236 : The author starts with the these histological findings (what does it indicate here)
  6. Section 2.1.4 Reprogramming need to be discussed at lines 265-266
  7. Line 393-394 It is important here the authors should discuss studies reported on these inhibitors
  8. Line 563 the author mentioned Warburg’s effects, could you explain here

Author Response

Firstly, we would like to thank the reviewer for her/his thorough reading of our manuscript and for her/his proposals to improve it. Accordingly, we have taken into account all comments and concerns. All changes are mentioned in the point-by-point answers below (italics), and are highlighted in yellow in the revised version of the manuscript.

In the current review, the authors have discussed repolarization of unbalanced MP as a therapeutic strategy to control chronic inflammation-related diseases such as Rheumatoid Arthritis, and cancer. Generally, monocytes and the tissue counterpart macrophages (MP) constitute the front line of the immune system. They can activate the immune system and monitor the immune response based on the signaling received. The balance between these dual activities is well managed and any type of unbalanced response of MP leads to pathological disorders. The authors suggested that repolarization is an important phenomenon as a therapeutic strategy.  

Decision: Minor comments

The paper was nicely written and needs some minor corrections before the final publication of the manuscript. Below are the comments for this paper to be incorporated in the revised version of the manuscript. 

  1. Section 1.3 needed to explore more on this

According to the reviewer’s request, we have given more information in this section (addendum highlighted in yellow).

  1. Line 154 In my view the authors need to discuss TAM here at least 2-3 lines

A short discussion has been added in new lines 167 to 172 (highlighted in yellow). In this short discussion we have shown TME. Therefore, TME showed in new line 518 has been removed (highlighted in yellow).

  1. Line 158-59, authors discuss the vast majority of the studies but cite only one reference.

Regarding the different M2 subtypes, we have cited 5 references ([21] to [25], lines 163-165). Ref [27] is the one that discusses the relevance of these different subtypes between mice and humans. This point is clarified in new line 175.

  1. Line 163-164 need to discuss figure 3

A short discussion has been added in new lines 182 to 186 (highlighted in yellow).

  1. Line 236 : The author starts with the these histological findings (what does it indicate here)

In order to clarify this paragraph (now starting at line 260), prior in the text we have referred to “histological” (in new line 247) and to “histological studies …” (in new line 254) (highlighted in yellow).

  1. Section 2.1.4 Reprogramming need to be discussed at lines 265-266

We have added one sentence (new lines 289 and 290, highlighted in yellow).

  1. Line 393-394 It is important here the authors should discuss studies reported on these inhibitors

According to the reviewer’s request, we have discussed the relevant studies. All new sentences are highlighted in yellow between new lines 418 and 439. Six supplemental references have been added.

  1. Line 563 the author mentioned Warburg’s effects, could you explain here

The Warburg’s effect is shortly explained in two sentences in new lines 600 to 603 (highlighted in yellow).
